# Parents of Children Diagnosed with Congenital Anomalies or Cerebral Palsy: Identifying Needs in Interaction with Healthcare Services

**DOI:** 10.3390/children10061051

**Published:** 2023-06-12

**Authors:** Ana João Santos, Paula Braz, Teresa Folha, Ausenda Machado, Carlos Matias-Dias

**Affiliations:** 1Department of Epidemiology, National Health Institute Doutor Ricardo Jorge, 1649-016 Lisboa, Portugal; paula.braz@insa.min-saude.pt (P.B.); m.teresa.folha@insa.min-saude.pt (T.F.); ausenda.machado@insa.min-saude.pt (A.M.); carlos.dias@insa.min-saude.pt (C.M.-D.); 2Comprehensive Health Research Center, National School of Public Health (ENSP), University of Lisbon, 1600-560 Lisboa, Portugal

**Keywords:** congenital anomaly, cerebral palsy, parents, healthcare services, healthcare professionals

## Abstract

The changes deriving from the birth of a child with a congenital anomaly (CA) or cerebral palsy (CP) imply, in many cases, an increased interaction with health services. A cross-sectional descriptive study was conducted with a convenience sample of parents of children diagnosed with four groups of CA (severe heart anomalies, spina bifida, orofacial clefts, and Down syndrome) and/or CP. A semistructured online questionnaire to be answered by parents was sent by web link to focal points of five parent associations and professional institutions. Data were analyzed through thematic content analysis (open-ended questions) and descriptive analysis (closed-ended questions). The results indicate consistency of responses of parents of children diagnosed with different conditions, namely with respect to the perception of health services and professionals. Closed and open-ended responses indicated three main topics in the interaction between health services and parenthood: information, coordinated and integrated responses, and support. The less positive outcomes suggest unmet information needs, while positive aspects include confidence in the care provided and the “training” received from health professionals.

## 1. Introduction

The birth of a child with a health condition or disability is known to impact families in a variety of ways [1,2,3]. When compared to control parents of healthy children, parents of children with disabilities or with chronic disorders have been found to experience greater general and psychological distress [4,5,6,7]. These children and families usually require health care or related therapeutic services more frequently than non-disabled children or children without chronic medical conditions [1]. Children’s healthcare needs and interactions with health services are associated with higher psychological and emotional demands on parents [1,8,9].

Studies have indicated that parents of children with disabilities or chronic long illnesses adjust to their child’s conditions, for the most part. Some may experience ongoing distress that can be associated with poorer health outcomes for parents, the child, and other family members [10]. Parental adjustment is related to health service interaction, including diagnosis transmission and communication with parents [1,5,11,12,13]. Parental adaptation is not only important for families. Evidence also suggests that when parents are engaged partners with healthcare providers in decision making and report satisfaction with provided services, child health outcomes improve and healthcare costs are reduced [4,8,13,14,15,16].

Congenital anomalies (CAs) are structural defects (congenital malformations, deformations, disruptions, and dysplasias) and chromosomal abnormalities [17]. Cerebral palsy (CP) is a term for a range of permanent movement difficulties caused by a non-progressive injury to the immature brain [18]. These conditions present varying prevalence, with 2 or 3% of children born with at least one CA [17] and around 2 per 1000 live births diagnosed with CP [18]. While these conditions have different rates of infant mortality, childhood morbidity, and long-term disability, all imply an increased interaction with healthcare services [17,18]. Families of children with health or developmental conditions may face similar challenges when interacting with healthcare services [1,5,10,19,20].

The aim of this study was to explore the interaction between families of children born with CAs or with cerebral palsy CP and healthcare services in Portugal.

The Portuguese health system features both private and public health sectors, the latter offering universal health coverage through the National Health Service (*Serviço Nacional de Saúde, SNS*). Public health care is shared between the central and regional governments, and both the public and the private sectors provide hospital care and community health services [21]. Public or public–private partnership health services, including primary care services and hospitals, tend to be the main providers of health services [22,23]. Data from 2017 indicate that approximately 25% of the population was covered by a health subsystem or voluntary health scheme [21]. Nonetheless, in 2019, out-of-pocket (OPP) spending corresponded to one of the highest shares among EU countries (OCDE) [23]. Unmet needs and limitations in terms of accessibility for subgroups of the population [23] have been recognized, namely for children with chronic diseases and deficiencies [24,25].

The aim of this study is to capture parental perception of the parenthood dimension in its interface with healthcare services, as well as common identified needs and gaps in healthcare provision.

## 2. Materials and Methods

A cross-sectional descriptive study was conducted with a convenience sample of parents of children diagnosed with four groups of CAs (down syndrome (DS), cleft lip with or without cleft palate (CLP), congenital heart defects (CHDs), and spina bifida (SB) and/or cerebral palsy (CP). A semistructured online questionnaire to be answered by parents was sent by web link to focal points (board members) of five parent associations and professional institutions in Portugal (Down Syndrome Parents Association, Spina Bifida and Hydrocephalus Association of Portugal, Happy Heart—Association for Protection and Support to Children with Heart Disease, the Portuguese Association of Friends of Children with Cleft Lip and Palate, the Federation of Cerebral Palsy Associations, and the Calouste Gulbenkian Cerebral Palsy Rehabilitation Center). Between 5 May and 27 May 2018, the questionnaire was disseminated by the focal points to families by email or through association websites and/or Facebook pages.

### 2.1. Instrument

A semistructured questionnaire was developed based on a literature review. The questionnaire addressed four thematic areas: diagnostic information and communication, parenthood, interaction with health services and professionals, and research. Each of the sections began with an open-ended question, followed by closed-ended items. We also assessed parental demographics, including sex, age, civil status, schooling, and region.

This paper reports part of the results of the parenthood section (1 question) and the interaction with health services and professionals (16 questions).

The section on parenthood included an open-ended question: “What do you feel would help you to play your role as a parent more effectively?”. The section titled Interaction with Health Services and professionals also began with an open-ended question (“What were the main problems when dealing with the health services?”), followed by 15 closed-ended items. Eight of these items were based on the Needs of Parents Questionnaire (NPQ). The NPQ was originally developed by Kristjánsdóttir [14,15] and consists of 51 statements that comprise a panorama of needs that parents might experience during a child’s hospital admission. Nine statements were retrieved from the NPQ instrument. As we also wanted to assess parents’ perceived interaction within outpatient services, three of these statements were changed slightly to make them relevant (for instance, from “staff” or “nurse” to “health professionals”). Additionally, we conducted meetings with the focal points from the parent associations about the most relevant topics for parents when interacting with health services and professionals. From the main areas identified, six statements were added: overall multidisciplinary health services and after the age of 18 years old, specialized care, communication between professionals, use of sensitive language, and psychological support.

The statements were assessed on the basis of two of the three different perspectives included in the NPQ instrument: (i) the parents’ perceived importance of each statement in relation to their child’s use of healthcare services (importance score) and (ii) fulfilment of the needs by healthcare services statements (fulfilment score). Each item was scored twice by parents. The importance score was examined with a five-point Likert scale ranging from “does not concern me” to “very important”, and the fulfilment score was examined with a five-point scale from “fully” to “not at all”.

The items of the NPQ were obtained from the Portuguese translated and validated version [26]. The overall version was sent to the focal points of the parent and professional associations, as well as to health professionals working with parents. Discrepancies and difficulties were reviewed and amended in accordance with received feedback.

We used RedCap 10.9.2 software to develop the online version of the questionnaire and generate a public link to be sent by the association focal points to parents.

### 2.2. Data Analysis

Descriptive statistical analysis, relative frequencies, and central tendency statistics were used for the closed-ended items. For analysis purposes, in the closed-ended questions of the interaction with services section, the relative frequency of the highest category of importance (“very important”) and fulfilment (“fully”) is described for the different statements. Cronbach’s alpha ranged from 0.93 on the importance scale and from 0.88 for the fulfilment scale.

The data from the two open-ended questions were analyzed through the method of thematic content analysis, given that it allows “qualitative data reduction and sense-making effort that takes a volume of qualitative material and attempts to identify core consistencies and meanings” [27] (p. 453). The analysis included several steps. All responses for each of the questions were read independently for overall impression, and one researcher coded the data. The units of analysis were defined throughout the coding process. In the first moment, the categories were developed based on the consensual themes (e.g., information and support) and repeatedly assessed against the empirical material. Two researchers further explored all text segments in each category with more in-depth categorical and theoretical–substantive coding categories, and additional categories were developed or modified whenever necessary, including by adding subcategories.

A researcher triangulation process [28] was implemented to further validate the results. A selection of 41 random units (uncoded segments) was provided to another researcher, in addition to the coding tree and category descriptions. The researcher coded each of the provided segments to the identified categories. The coding was compared, and the results indicated a higher number of discrepancies for the parenthood open-ended question in comparison to the health services question. These discrepancies were mainly related to the subcategories; case consensus was mainly achieved by changing category descriptions. In very few cases, the statement was coded differently. No software was used. The final coding tree developed through open coding corresponded to the consensus among the members of the research team [28].

The sections varied in terms of completeness. For analysis purposes, each section was considered independently, and completed responses were considered for the different sections.

## 3. Results

Overall, 254 questionnaires were returned, with 73% (n = 186) completed for all the sections. The majority of participants (74%) received the questionnaire through a parent or patient association: the Down syndrome Parents Association (27%), cerebral palsy institutions (23%), the Cleft Lip and Palate Association (16%), or the Spina Bifida and Hydrocephalus Association (8%). The remaining questionnaires (26%) were received from other institutions or professionals (6%). 

Most of the respondents were women (88.7% vs. men 11.3% men), married or living in civil partnership (79.6%); between 35 and 44 (49%) and 45 and 54 years of age (28%) and had up to 12 years of schooling (secondary school) (66%).

### 3.1. Parenthood

From the total of completed responses in the parenthood section (n = 186), the thematic content analysis yielded 7 categories and 10 subcategories (see Table 1): (1) support, (2) information, (3) resources and financial support, (4) time and availability, (5) health system and services, (6) inclusive society, and (7) rehabilitation and development.

Most of the categories related to society sectors and services, representing an external dimension of parenting that influences the upbringing of children with disabilities or specific health conditions. Some categories related to parents’ and families’ needs, while others are linked to child development.

The need for support (1. Support) and for better and more precise information (2. Information) seemed to be expressed by a large number of respondents. Parents indicated a lack of support for families and parents, primarily from the government and community (1. Support). Lack of support was widely expressed in overall generic terms (e.g., “more support”; “further responses”; “direct support”), general statements about specific societal sectors, such as the state, social spheres, and community (e.g., “more state support”; “support from the state in all levels”; “appropriate social support”), and expression of families’ needs for support (e.g., “more support to families”; “support for parents”; “more responses for us [parents]”).

Information pertaining the child’s conditions was seen as essential and as missing in the different areas that influence the child’s life (health, services, disability rights, and education). As in the support category, this perception was mostly expressed in general terms, qualifying not only the amount (e.g., “information”; “more information”; “lack of information”) but also the quality of the information (e.g., “more precise information”; “reliable information”; “more coherence in the information provided”).

In addition to the statements more commonly observed in the first two categories, parents also stated specific areas that could benefit families of children with the abovementioned conditions, including financial support (3. financial resources and financial support), time, and flexible work hours (4. time and availability).

Fewer responses highlighted the child’s needs, specifically the need for early developmental intervention (7. rehabilitation and development) and an inclusive society (6. inclusive society). On the topic of health (5. health system and services), the statements indicate the need for improvement in the provision of services, as well as in the information and knowledge of professionals about the specific conditions of the children. This perception also comes across in the second most frequent category (2. information), where parents expressed the need for better-informed professionals and more information about the specific and sometimes rare conditions of their children, as well as acute care (2.2. health).

### 3.2. Interaction with Health Services and Professionals

The section pertaining to interaction with health services and professionals comprised 145 completed responses. Thematic content analysis of the open-ended question, “What were the main problems when dealing with the health care services?” yielded five categories and four subcategories (see Table 2): (1) access to healthcare services, (2) information and knowledge, (3) specialized resources, (4) sensitivity and lack of empathy and support, and (5) coordinated and integrated response.

Most categories were related to objective aspects within the health services, namely barriers to access services and professionals (1. access to healthcare services), limited transmitted information and knowledge of the child’s conditions by some professionals (2. information and knowledge), lack of specialized human or technical resources (3. specialized resources), and lack of coordination between services and between professionals to provide integrated responses (4. coordinated and integrated response). One category (4. sensitivity and lack of empathy and support) referred to parents’ perceptions about how services and some professionals dealt with parents and the family of children with health and developmental conditions. These segments not only included narratives of family’s experiences but also overall feeling about anxiety and difficulties in being overlooked and dismissed.

The 15 closed items focused on the information that was provided to parents, satisfaction with the treatment or care provided to the child, quality and trust in health services, and aspects related to the attitudes and behavior of health professionals (Figure 1). All the assessed aspects resulting from the interaction with health services and professionals were perceived as highly important by the parents (>88%).

Marked differences were observed as to whether these aspects are (or not) considered a reality in the healthcare services. The highest discrepancy (between the percentage of respondents who say it is “very important” and the percentage of those who perceive it as “a reality in health care services”) was observed for items pertaining to psychological support for parents (87.5% vs. 40.7%), receiving written information (97.6% vs. 51.2%), and being encouraged to ask questions (96.7% vs. 42.7%). For other items assessing information about specific issues, the discrepancy was lower. Most participants did report to believe that the health services and professionals fulfilled these, namely to teach them how to care for their child (81.6%), to be informed about what affects the child’s development (74.4%), and to be informed of all known health outcomes (68.7%). Overall, a large portion of respondents felt that they were an integrated part of the health care provided to their child. Three-fourths (75.8%) reported that they were able to make the final decision when it came to treatment of their child, and 70.4% felt their own knowledge about the child was taken into consideration. About two-thirds also felt professionals recognized and considered parents’ feelings (60.3%).

## 4. Discussion

The results of this study of parents of children diagnosed with Down syndrome (DS), cleft lip with or without cleft palate (CLP), severe congenital heart defects (CHDs), spina bifida (SB), or cerebral palsy (CP) cover two topics: parenthood and interaction with health services and professionals. This study focused solely on parents’ perceptions, using a convenience sample implemented through a qualitative approach. Hence, caution is required in generalizing the findings.

One major outcome is the homogeneity of the results for parents of children diagnosed with different conditions. All these conditions, some of them rare, might warrant different types of health and therapeutic interventions extended for different periods of time and with very distinct outcomes [29,30,31,32,33]. Parents of children diagnosed with these five conditions provided remarkably similar responses on issues about parenthood and on the interaction with healthcare services. Three themes were found in both parenthood and interaction with health services: information, coordinated and integrated responses, and support.

Information was an emergent common theme in all analyzed survey sections. Information was the second most frequent category in both open-ended questions. Parents felt that their role would improve upon receiving information, either overall information or topic specific to their child’s life. The results of open and close-ended questions express opposing perceptions of information when interacting with healthcare services. Information was seen as a major issue when dealing with health services, and the closed-ended questions showed that less than half of the parents felt they could ask questions or had received written information. Moreover, parents positively assessed some aspects of information provided within healthcare services: to be informed about what affects their child’s development, of the known health outcomes, and to learn how to care for their children. One possible explanation for these unmet information needs is parents’ expectations and different perceived outcomes of communication between parents and healthcare professionals. Studies have indicated that information is an important part of parental and familial adjustment [34,35,36,37], representing not only the need for understanding of the health condition, future impacts, and care but also providing a sense of control and satisfying the need for reassurance [9,34,38,39,40]. Hence, parents might have felt that they would require more information, even though additional information could not be provided at the time or in the way parents would like to have received. Parents may need more time, different types of communication, and materials to be able to process and integrate the provided information and knowledge [11,37,39]. The importance of considering information not as a single exchange but rather as a process that needs to be adapted and revisited is commonly acknowledged in studies focusing on CAs and CP diagnosis [11,37,41,42,43]. In addition, it is important to note that time of the diagnostic communication impacts parents’ perception, as prenatal diagnosis may help parents to integrate the child’s condition and provide additional time and space for information communication [2,36,37]. Early detection, such as recent published data regarding early signs in the first trimester that predict the presence of spina bifida or CHD [44,45], may have a significant impact on parental adjustment. 

Another potential perspective relates to the degree of specialization of the professionals. Families of children with these conditions will probably have high health care use, even if at different rates and during different periods of time [1,3,10]. It may be expected that parents interact with different health professionals and services, including those that are not specialized in their children’s condition. While there might be a high level of trust in healthcare workers, with respect to information and care [46], the lack of knowledge of general healthcare workers about specific conditions [42,43] can be, in parents’ perception, detrimental. In this study, parents perceived a lack of information and a lack of professional knowledge and training about the specific and sometimes rare conditions of their child. More highly resourced centers can better promote contact with medical specialists may help to deliver more personalized information [46]. It has been suggested that in Portugal, health service delivery for children could benefit from the training of pediatricians and general practitioners in the diagnosis and management of disabling conditions [24], defining a network of highly specialized pediatric centers of competence [25].

This relates to the second important result: coordinated and integrated responses. 

The need for multidisciplinary responses that articulate between one another when dealing with health services was clearly indicated in both open-ended questions. This was also identified in the needs assessment closed items, where only 59% of parents felt professionals coordinated and talked to one another. A sense of the system’s fragmentation and arbitrariness has been previously identified as a barrier to healthcare use among parents of children with vulnerabilities [47]. Care coordination is not only important for families and parents but promotes strong partnerships between families and physicians and supports shared medical decision making [1], with positive health outcomes [48]. It is important to consider whether the current system of health care may emphasize acute illness and well-child care, with detrimental and visible impacts on the long-term management of chronic conditions and disabilities [1].

Finally, support was another key aspect mentioned by parents in relation to parenthood performance. One of the problems parents indicated to have emerged when interacting with healthcare services was a lack of sensitivity and empathy or support from services and professionals, even though this was not a very frequent category. In the closed-ended items, parents stated that they did not feel that professionals employed accurate and sensitive descriptions for their child’s condition. Communication with parents may not only be affected by language, culture, ethnicity, income, and education but also by the sensitivity of the discourse adopted by professionals [2,13,33,42].

The study is not without limitations, the first being the fact that a convenience sample was used. Parents were recruited from contact lists of Portuguese parent and patient associations. These parents may, therefore, be more active and involved or have different reflections of health care services when it comes to their children’s needs. In addition, information about the children was not included, and different perceptions of parents could have also originated from children with different conditions and varying degrees of disabilities. However, the overall perception of parents contributes to the exploration of the impressions of parents of children with chronic or long-term health conditions. Other than the convenience sample, the qualitative data analysis reflects a general schema inferred from the analysis of participants’ responses and may be acquiescent to interpretation. The process used to validate the analysis may enhance trustworthiness, but the focus on these parents’ perspectives may need further translation for actual use in healthcare settings and would benefit from comparison with the perspectives of health professionals.

Possibly related to the high level of interaction of these children with healthcare services, the parents’ perceptions of their role was intertwined with health services. Information, coordinated and integrated responses, and support were three subjects similarly found in the interaction between health services and parenthood. The less positive responses suggest unmet information needs, while positive aspects included confidence in the care provided and in the “training” received from health professionals. These results, along with the perceived lack of collaboration between professionals and lack of emotional and psychological support, indicate that for parents, health care is also defined by psychosocial experiences. Although this study did not focus on assessing service delivery, it seems important to explore whether, in Portugal, healthcare services for children with chronic or complex health conditions could be delivered through a more client-centered approach to tackle these aspects.

## Figures and Tables

**Figure 1 children-10-01051-f001:**
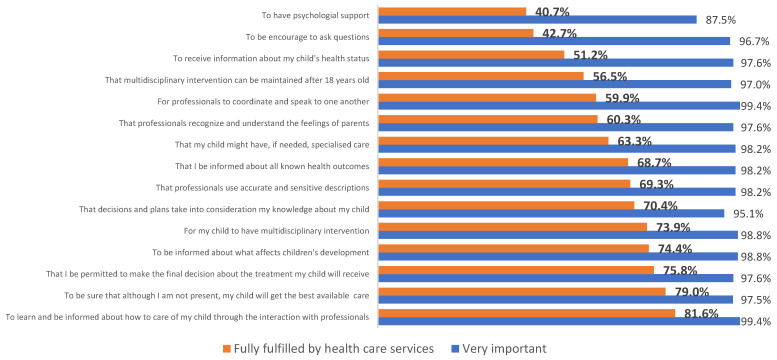
Distribution of importance for parents and fulfilment by healthcare services of 15 statements assessing aspects pertaining to interaction with health services and professionals’.

**Table 1 children-10-01051-t001:** Coding tree of the open-ended question, “What do you feel would help you to play your role as a parent more effectively?”.

Category/Subcategory	Description	No. of Segments	Examples
1. Support	Category including statements about the need for support, divided into three subcategories that describe overall support and support with respect to specific society sectors or specific areas.	71	
1.1. Overall support	General references to the need for more and better support, without specifying areas or types of support.	24	*“social support”*
1.2. State, social, and community support	Need for more and/or better support either from the state, from social services, or from the community.	28	*“further assistance by the state”*
1.3. Family and caregiver support	Need for help, support, and/or services for the family and caregiver.	19	*“extra support to the families”*
2. Information	Category including statements and reports of participant needs for information, divided into four subcategories that describe need for overall information and in specific sectors and services.	54	
2.1. Overall information	Qualification of the overall needed information, such as as further, better, coherent, and/or more adequate information.	19	*“more… better… information”*
2.2. Health	Better informed professionals and more information on the condition and acute care.	17	*“reliable information from health professionals”*
2.3. Services and rights	Information on the available services and disability rights.	10	*“Information on rights of disabled* *individuals”.*
2.4. Education	Information on the schooling system, as well as education techniques.	8	“*Information on schooling and teaching techniques*”
3. Financial resources and financial support	Category including statements and reports about financial needs.	49	*“financial help to buy technical assistance technology”*
4. Time and availability	Category including statements about parents needing more time and being more available to respond to their children’s needs.	47	*“time and availability for the medical appointments and therapies”*
5. Health system and services	Category including statements and reports of participants about what they found lacking in the healthcare services provided, divided into three categories: more or better overall services, capable health services, and professionals.	39	
5.1. Overall	General references to the need for better and more services.	15	*“better health services”*
5.2. Integrated and transdisciplinary care	Indication of the lacking qualities that could improve the provided care: accessibility, transdisciplinarity, and timely response.	17	*“more accessible and more emphatic health services, with less bureaucracy and more available schedules”*
5.3. Professionals	Statement about support and know-how of the professionals on the specificities of the condition.	7	*“physicians with increased know-how”*
6. Inclusive society	Category including concern expressed by paretns about a more inclusive community and society.	24	*“true integration”*
7. Rehabilitation and development	Category including statements and reports about the importance and overlooked need for early rehabilitation and interventions to promote several areas of child development.	15	*“timely responses to promote my sons development”*

**Table 2 children-10-01051-t002:** Coding tree of the open-ended question, “What were the main problems when dealing with the health care services?”.

Category/Subcategory	Description	No. of Segments	Examples
1. Access to healthcare services	Category including statements and reports about difficulty in accessing services and care provision due to specific logistic and functioning issues or in contacting health professionals.	69	
1.1. Services and care provision	Overall difficulties in accessing specific services or treatments related to timely responses, procedures, and bureaucracy.	54	*“he was referred to the hospital for a development appointment. […] A year later we were called to our first hospital visit”*
1.2. Professionals	Difficulties in accessing and contacting professionals.	15	*“Difficulties (sometimes total impossibility) of contact when dealing with professionals from the NHS”*
2. Information and knowledge	Category including statements about the lack of overall information and lack of professional knowledge and training.	61	*“lack of information”* *“A considerable degree of ignorance on the condition”*
3. Specialized resources	Category including statements and reports about the lack of specialized resources, either human, technical, or services.	38	*“shortage of specialized medical staff”* *“Impossible to be referred to specialized services”*
4. Sensitivity and lack of empathy and support	Category including statements and reports about perceived lack of sensitivity and empathy or support from service providers and professionals.	32	*“a lack of empathy towards our anxiety”* *“no psychological support to us, the family”*
5. Coordinated and integrated response	Category including statements about the lack of coordination between professionals and services, as well the lack of flexibility of healthcare services to respond to different needs.	27	
5.1. Coordination between services and professionals	Difficulties resulting from a lack of communication and coordination between services and care providers.	19	*“lack of coordination and collaboration among physicians of various specialties”*
5.2. Integrated and transdisciplinary care	Lack of an integrated and transdisciplinary health provision, fragmented in several, sometimes distinct responses.	8	*“different procedures, the failure of transdisciplinary”*

## Data Availability

The data presented in this study are available upon request from the corresponding author. The data are not publicly available due to privacy and ethical restrictions.

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
