# Peer review of "Parents of Children Diagnosed with Congenital Anomalies or Cerebral Palsy: Identifying Needs in Interaction with Healthcare Services"

_children, 2023, doi:10.3390/children10061051_

Round 1

Reviewer 1 Report

Dear authors, as an Orthopaedic Pediatric surgeon, dealing with families with CP and limb deformities, I enjoyed your paper, even that it was hard for me to follow our your statements.

Your remarks regarding the interface between parents and health services are important.  Please, in your remarks, you must distinguish among parents of children with less severe CP ( hemiplegia) that their children are walkers indepentenlty and require more social and financial care in order to achieve good results in school and sport activities and those with severe CP (quadroplegia) that require expensive instruments ( chairs for spine deformities, parawalkers). Their demands are mainly social care of their children and specialized schools. 
 Parents ask for specialized information ( you mention on line 253), but it is difficult for every child to attend a specialized clinic, where surgical treatment will take more time, than when dealing in their local hospital.  Parents EXPECTATIONS often are HIGH, they blame health professionals, when results are not similar to their expectations. This part was NOT considered in your analysis.

Your results for the common needs of families of disabled children are IMPORTANT.

Author Response

Response:

Thank you for all the comments. We have revised the manuscript, included information and rewritten the manuscript to respond. Please find bellow a more detailed description of the revisions.

R1.1: “Your remarks regarding the interface between parents and health services are important.  Please, in your remarks, you must distinguish among parents of children with less severe CP ( hemiplegia) that their children are walkers indepentenlty and require more social and financial care in order to achieve good results in school and sport activities and those with severe CP (quadroplegia) that require expensive instruments ( chairs for spine deformities, parawalkers). Their demands are mainly social care of their children and specialized schools.”

Response: although this a very important issue, the questionnaire did not include information regarding the child. The reasoning being to be better received by parents and because our focus was not to characterise the health services deliver, but the perception of parents of that interaction. In regard to this comment and others from other reviewers the limitations were rewritten (Lines 326-333):

“The study is not without limitations, the first being the fact that a convenience sample was used. Parents were recruited from contact list of parents within Portuguese parents and patients associations. These parents may, therefore, be more active, involve or have a different reflexion of health care services, when it comes to their children needs. In addition, the children information was not included and different perceptions from parents could also originate from children with different conditions and disabilities degrees. However, the overall perception of parents contributes to explore the impression of parents of children with chronic or long-term health conditions.”

R1.2.: “Parents ask for specialized information (you mention on line 253), but it is difficult for every child to attend a specialized clinic, where surgical treatment will take more time, than when dealing in their local hospital.  Parents EXPECTATIONS often are HIGH, they blame health professionals, when results are not similar to their expectations. This part was NOT considered in your analysis.”

Response: thank you for this comment. As the study focus on the parents’ perception, it is important to keep in mind that these are parents perceptions and that they may have not feasible expectations. The discussion was rewritten to take into account this aspect (Lines 274-305):

“One possible explanation for these information unmet needs could be parent’s expectations and different perceived outcomes of communication between parents and health care professionals. Studies have indicated that information is an important part of the parents and family’s adjustment [35]–[38]. It does represent not only the need for understanding of the health condition, future impacts and care, but also provides a sense of control and the need for reassurance [9], [35], [39]–[41]. Hence, parents might have felt that they would require more information, even though additional information was not possible to be provided at the time or in the way parents would like to have received. Parents may need more time; different types of communication means and materials to be able to process and integrate the provided information and knowledge [11], [38], [40]. The importance of considering information not as a single exchange, but rather as a process that needs to be adapted and revisited is commonly acknowledge in studies focusing on CAs and CP diagnosis [11], [38], [42]–[44]. In addition, it is important to note that time of the diagnose communication impacts parents perception, as prenatal diagnosis may help parents to integrate the child’s condition and provide additional time and space for information communication [37], [45], [46]. Early detection, as recent published data regarding early signs in the first trimester that predict the presence of spina bifida or CHD [47], [48], may have a significant impact for parents adjustment.

Other potential perspective relates to the degree of specialization of the professionals. These families will probably have high health care use, even if at different rates and during different periods of time [1], [3], [10]. It may be expected that parents interact with different health professionals and services, including those that are not specialized in their children’s condition. And while, there might be a high level of trust in health care workers, as information and care is concerned [49], the lack of knowledge from general health care workers on specific conditions[43], [44] can be, in parents perception, detrimental. In this study, parents perceived a lack of information and professional’s lack of knowledge and training about the specific and sometimes rare conditions of their child. More highly resourced centers can better promote contact with medical specialists may help to deliver more personal in information [49]. It has been suggested that, in Portugal, the health service delivery for children could benefit from the training of paediatricians and General Practitioners in the diagnosis and management of disabling conditions[25], and the defining a network of highly specialized pediatric centers of competence[26].”

Reviewer 2 Report

Authors present an interesting survey on the parents’ efforts and needs in taking care of children with cromosomal or congenital anomalies. I would suggest adding a paragraph in the Discussion highlighting how prenatal diagnosis is nowadays anticipating some of these diagnoses. There are recent published data regarding early signs in the first trimester that predict the presence of spina bifida (PMID: 31875250) or CHD (PMID: 36468264). These may increase in the future the parents self-consciousness of the affected conditions and may help them to be better prepared to face the difficulties in managing the assistance to their children.

Page 2 Line 55: “the” without capital letters

Please check minor English mistakes throughout the manuscript

Author Response

Thank you for all the comments. According to yours’ and other reviewer’s comments, we have revised the manuscript, included information and rewritten the manuscript to respond (all marked in yellow).

We have included a paragraph in the discussion highlighting the importance of pre-natal diagnose (Lines 286-291):

“In addition, it is important to note that time of the diagnose communication impacts parents perception, as prenatal diagnosis may help parents to integrate the child’s condition and provide additional time and space for information communication [37], [45], [46]. Early detection, as recent published data regarding early signs in the first trimester that predict the presence of spina bifida or CHD [47], [48], may have a significant impact for parents adjustment.”

Reviewer 3 Report

This article highlights aspects of patient and family-centred care that are crucial to the wellbeing of children with a CA or CP and their families and that go beyond physical health.

The introduction would benefit from some description of how the healthcare system is organised in Portugal (where I assume the families accessed services), this is relevant given that the authors allude to "the organisation structure and characteristics of health care services" (line 267) limiting their ability to be responsive to parents' needs of information. Different countries face different challenges regarding healthcare organisation; whilst the study focuses on parents' identified needs and interactions with professionals, the authors make organisation of the healthcare system key to their discussion and concluding argument. Following from this, I was surprised to find the acronym "NHS" in figure 1 without a description of what it stands for, and I found it confusing given that the UK national system is globally known as NHS. Is this referring to the Portuguese national health system?

The lack of transparency regarding demographics greatly reduces the quality of this article. For example, in their discussion, the authors argue homogeneity of responses is a major outcome and make reference to the range in children's ages (line 241), yet there is no demographic information to assess the variety of ages. The survey contains one item "That a multidisciplinary intervention can be maintained, after my childs 18 years", parents are asked to assess perception and fulfilment. It is unclear whether families have had the experience of their child turning 18 so as to be able to evaluate the continuity of services.

More information is needed to substantiate the key claim that "The less positive outtakes seemed to convey a health service system that is more acute-illness oriented and not as much focused on long-term management of chronic conditions, with fragmented service delivery and lack of coordination" (lines 304 to 306). Healthcare systems are complex, and sadly, this article takes a light-touch approach to analyse interactions between patients and the system. Without evidence emerging from the data, and consideration of structural factors, this conclusion seems to be based on an assumption rather than evidence.

Minor grammatical errors

Author Response

Thank you for all the comments. We have revised the manuscript, included information and rewritten parts of the manuscript to respond. Please find bellow a more detailed description of the revisions (marked in yellow in the revised version).

R3.1.: “The introduction would benefit from some description of how the healthcare system is organised in Portugal”

Response: The following text was added to the introduction between the lines 57 and 68.

“Portugal Health System features both a private and public health sector, the latest offering universal health coverage through the National Health Service (Serviço Nacional de Saúde, SNS). Public health care is shared between the central and regional governments, and both the public and the private sectors provide hospital care and community health services [22]. Public or public-private partnership health services, including primary care services and hospitals tend to be the main providers of health services [23], [24]. Data from 2017, indicated that approximately 25% of the population was covered by a health subsystem or voluntary health scheme [22]. Nonetheless, in 2019 Out Of Pocket (OPP) spending corresponded to one of the highest shares among EU countries (OCDE) [24]. Unmet needs and limitations in accessibility for subgroups of the population[24], have been recognised, namely, for children with chronic diseases and deficiencies [25], [26].”

R3.2. “Following from this, I was surprised to find the acronym "NHS" in figure 1 without a description of what it stands for, and I found it confusing given that the UK national system is globally known as NHS. Is this referring to the Portuguese national health system?”

Response: National Health Service corresponds to translation of the Portuguese “Serviço Nacional de Saúde”, commonly used in English publications. It was actually inspired by the British model. However, as the questions addressed health care services within Portuguese health system (and not only public health care) the legend in the figure was corrected in accordance.

R3.3 “The lack of transparency regarding demographics greatly reduces the quality of this article. For example, in their discussion, the authors argue homogeneity of responses is a major outcome and make reference to the range in children's ages (line 241), yet there is no demographic information to assess the variety of ages.”

Response: Data on the parent’s demographics is only described in text and no additional table was added, considering the limit number of tables/figures and words. The questionnaire did not include information regarding the child. The reasoning being to be better received by parents and because our focus was not to characterise the health services deliver, but the perception of parents of that interaction. The homogeneity of responses still seems to us a major outcome, as parents from children with different conditions stated similar narratives when interacting with health services. However, we agree that as the demographics of the child have not been accessed this limits our findings. About this and other similar comments the discussion has been rewritten (please see bellow, R.3.5.).

R.3.4. “The survey contains one item "That a multidisciplinary intervention can be maintained, after my childs 18 years", parents are asked to assess perception and fulfilment. It is unclear whether families have had the experience of their child turning 18 so as to be able to evaluate the continuity of services.”

Response: As stated in the methods section, part of the close-ended items derived from the meetings with the focal points of the parents associations. In the specialised care and services topic one of the main concerns brought up by the parents was the discontinuation of specialised health care services after the child reaches 18 years old. Hence, the item was included in the questionnaire. We have rewritten part of the methods section regarding the instrument to make this clearer (Lines 101-110):

“Nine statements were retrieved from the NPQ instrument. As we also wanted to assess parents’ perceived interaction within outpatient services, three of these statements were changed slightly to make it relevant (for instance, from “staff” or “nurse” to “health professionals”). Additionally, we conducted meetings with the focal points from the parents associations about the most relevant topics for parents when interacting with health services and professionals. From the main areas identified, six statements were added: multidisciplinary health services overall, and after the age of 18 years old, specialized care, communication between professionals, use of sensitive language, and psychological support.”

R3.5: “More information is needed to substantiate the key claim that "The less positive outtakes seemed to convey a health service system that is more acute-illness oriented and not as much focused on long-term management of chronic conditions, with fragmented service delivery and lack of coordination" (lines 304 to 306). Healthcare systems are complex, and sadly, this article takes a light-touch approach to analyse interactions between patients and the system. Without evidence emerging from the data, and consideration of structural factors, this conclusion seems to be based on an assumption rather than evidence.”

Response: The discussion was rewritten to account for this and previous comments:

Lines 253-255: “. The study focused solely on parent’s perceptions, using a convenience sample and included a qualitative approach. Hence, caution is needed in generalizing the findings.”

Lines 267-305: “The results from open and close-ended questions express opposing perceptions of information when interacting with health care services. Information was seen as a major issue when dealing with health services and the closed-ended questions showed that less than half of the parents felt they could ask questions or had received written in-formation. At the same time, parents positively assessed some aspects of information provided within health care services: to be informed about what affects their child development, of the known health outcomes, and about learning how to care for their children. One possible explanation for these information unmet needs could be parent’s expectations and different perceived outcomes of communication between parents and health care professionals. Studies have indicated that information is an important part of the parents and family’s adjustment [35]–[38]. It does represent not only the need for understanding of the health condition, future impacts and care, but also provides a sense of control and the need for reassurance [9], [35], [39]–[41]. Hence, parents might have felt that they would require more information, even though additional information was not possible to be provided at the time or in the way parents would like to have received. Parents may need more time; different types of commu-nication means and materials to be able to process and integrate the provided infor-mation and knowledge [11], [38], [40]. The importance of considering information not as a single exchange, but rather as a process that needs to be adapted and revisited is commonly acknowledge in studies focusing on CAs and CP diagnosis [11], [38], [42]–[44]. In addition, it is important to note that time of the diagnose communication impacts parents perception, as prenatal diagnosis may help parents to integrate the child’s condition and provide additional time and space for information communication [37], [45], [46]. Early detection, as recent published data regarding early signs in the first trimester that predict the presence of spina bifida or CHD [47], [48], may have a significant impact for parents’ adjustment.

Other potential perspective relates to the degree of specialization of the professionals. These families will probably have high health care use, even if at different rates and during different periods of time [1], [3], [10]. It may be expected that parents interact with different health professionals and services, including those that are not specialized in their children’s condition. And while, there might be a high level of trust in health care workers, as information and care is concerned [49], the lack of knowledge from general health care workers on specific conditions[43], [44] can be, in parents perception, detrimental. In this study, parents perceived a lack of information and professional’s lack of knowledge and training about the specific and sometimes rare conditions of their child. More highly resourced centers can better promote contact with medical specialists may help to deliver more personal in information [49]. It has been suggested that, in Portugal, the health service delivery for children could benefit from the training of paediatricians and General Practitioners in the diagnosis and management of disabling conditions[25], and the defining a network of highly specialized paediatric centers of competence[26].”

Lines 326-333: “The study is not without limitations, the first being the fact that a convenience sample was used. Parents were recruited from contact list of parents within Portuguese parents and patients associations. These parents may, therefore, be more active, involve or have a different reflexion of health care services, when it comes to their children needs. In addition, the children information was not included and different perceptions from parents could also originate from children with different conditions and degrees of disabilities. However, the overall perception of parents contributes to explore the impression of parents of children with chronic or long-term health conditions.”

Lines 340-350: “Information, coordinated and integrated responses and support were three subjects similarly found in the interaction between health services and parenthood. The less positive outtakes suggest unmet information needs, while positive aspects included confidence on the care provided and on the “training” received from health professionals. These results, along with the perceive lack of collaboration between professionals, and lack of emotional and psychological support, indicate that, for parents, health care is also defined by the psychosocial experiences. Even though, the study did not focus on assessing service delivery, it seems important to explore if, in Portugal, health care services for children with chronic or complex health conditions could manage a more client-centre approach that tackles these aspects.”

Round 2

Reviewer 3 Report

No additional comments

No additional comments